# CARE: a Benchmark Suite for the Classification and Retrieval of Enzymes

**Jason Yang**
Chemistry and Chemical Engineering
California Institute of Technology

**Ariane Mora**
Chemistry and Chemical Engineering
California Institute of Technology

**Shengchao Liu**
Computing and Mathematical Sciences
California Institute of Technology

**Bruce J. Wittmann**
Office of the Chief Scientific Officer
Microsoft Corporation

**Anima Anandkumar**
Computing and Mathematical Sciences
California Institute of Technology

**Frances H. Arnold**
Chemistry and Chemical Engineering
Biology and Biological Engineering
California Institute of Technology

**Yisong Yue** [*]
Computing and Mathematical Sciences
California Institute of Technology

## Abstract

Enzymes are important proteins that catalyze chemical reactions. In recent years, machine learning methods have emerged to predict enzyme function from sequence; however, there are no standardized benchmarks to evaluate these methods. We introduce CARE, a benchmark and dataset suite for the Classification And Retrieval of Enzymes (CARE). CARE centers on two tasks: (1) classification of a protein sequence by its enzyme commission (EC) number and (2) retrieval of an EC number given a chemical reaction. For each task, we design train-test splits to evaluate different kinds of out-of-distribution generalization that are relevant to real use cases. For the classification task, we provide baselines for state-of-the-art methods. Because the retrieval task has not been previously formalized, we propose a method called Contrastive Reaction-EnzymE Pretraining (CREEP) as one of the first baselines for this task and compare it to the recent method, CLIPZyme. CARE is available at `https://github.com/jsunn-y/CARE/`.

## 1 Introduction

Proteins, which are sequences of amino acid building blocks, are not only integral components of all living organisms, but also important for a myriad of commercial applications spanning from the health domain to the bio-economy. Enzymes are a subclass of proteins that can catalyze chemical reactions, and they have many applications in areas such as bioremediation, plastic degradation, gene editing, and drug synthesis [1, 2, 3, 4].

Identifying the specific chemical reactions that an enzyme is capable of performing (*i.e.* the enzyme's function) is a key first step for many applications [5]. While hundreds of millions of proteins

---

[*]Corespondance: yyue@caltech.edu

38th Conference on Neural Information Processing Systems (NeurIPS 2024) Track on Datasets and Benchmarks.

have been sequenced, less than 1% are annotated with function [6]. For instance, during standard metagenomic analyses, enzyme genes are annotated for their functions, which enables determination of an organism's critical metabolic pathways and specialization [7, 8, 9]. When applied to chemical synthesis, enzyme annotations are needed to identify catalysts that can replace existing step(s) in drug synthesis procedures–known as retrobiosynthesis [10]. Likewise, enzyme engineering for new-to-nature function [11] involves discovering an enzyme starting point with a desired function before improving activity for that function using protein engineering techniques [12, 13]. Historically, similarity search algorithms, most notably BLAST, have been the most common methods used to assign function to protein sequences [14, 15]. These methods work by finding similar sequences in annotated reference databases, as similar sequences are likely to share function. More recently, searching based on structural similarity (Foldseek) has also demonstrated utility for assigning function [16, 17]. However, up to one third of genes from bacterial genomes cannot be assigned function using existing methods [7]. As such, there is a need for more abundant, high-quality annotations of enzymes and automated workflows to identify enzymes with desired functions.

In recent years, there has been increasing interest in using machine learning (ML) for a broad range of applications related to the functional prediction and design of enzymes [13, 18, 19]. In particular, ML models have emerged to classify protein sequences based on their function, which are reviewed here [20] with a few examples listed here [21, 9, 22, 23, 24, 25, 26]. Despite these advances, there is no standard benchmark or dataset for evaluating computational models for enzyme function prediction [27]. A challenge associated with classification is that a given enzyme is often able to perform multiple reactions [28], and many reactions are not annotated. Moreover, complex tasks such as extrapolation to unannotated reactions [29] have yet to be evaluated.

In this work, we present a benchmark suite for the **C**lassification **A**nd **R**etrieval of **E**nzymes (CARE). Our contributions can be summarized as: **(1)** formalizing model evaluation into two tasks that encompass applications relevant to scientists and engineers: classification of an enzyme by function (Task 1), and retrieval of enzyme sequences based on a reaction (Task 2); **(2)** curating high-quality and easy-to-use datasets; **(3)** providing train-test splits that mimic challenging extrapolations in real-world use cases; and **(4)** benchmarking state-of-the-art models for Task 1, and providing a new method that serves as a baseline for Task 2. Because Task 2 has been minimally explored, we introduce a model called **C**ontrastive **R**eaction-**E**nzym**E** **P**retraining (CREEP) to serve as a baseline for text, reaction, and sequence integration and compare it to an existing approach (CLIPZyme) that uses sequence and reaction, in addition to other retrieval approaches. CREEP can perform contrastive learning across three different modalities (protein, reaction, and textual description), and the learned representations are then used for retrieval. Overall, we anticipate that CARE will be a useful and easy-to-use resource for ML researchers to benchmark their enzyme function prediction models.

## 2   Related Work

**Datasets.** Various databases have emerged to help researchers store, share, and identify functionally annotated enzymes. Protein sequence databases such as UniProt [6] and Pfam [30] are catalogs of annotated protein sequences. While most databases reference protein sequences, increasingly, these sequences can be linked to protein structures, either experimentally validated, as in the Protein Data Bank (PDB), or via structural prediction tools [31] in databases such as the AlphaFold Database [32]. BRENDA [33] is a curated database specific for reaction and enzyme sequence information. Rhea [34] consolidates information from BRENDA, and other sources, such as pathway databases including KEGG [35]. There is ongoing work to compile and clean/standardize reactions from multiple databases, namely ECReact [36] and EnzymeMap [37]. Related to these databases, there are retrobiosynthesis planning tools [38, 39, 40], and Selenzyme is tool to retrieve enzymes to perform a target reaction [41].

**Protein Benchmarks.** Our work takes inspiration from existing protein fitness prediction benchmarks, where fitness is a quantification of some function. TAPE [42] and FLIP [43] evaluate representations from protein language models for the prediction of a broad range of general and specific protein properties (stability, secondary structure, binding etc.) ProteinGym considers sequence variant effect prediction by using likelihoods from these language models [44]. While benchmarks for protein fitness prediction tasks are well defined, there are no standardized benchmarks for protein function prediction. Fitness is a numerical quantification of protein function (stability, enzyme activity

level, etc.), but function is more qualitative/categorical, e.g. a description of an enzymatic reaction associated with a protein. Here, we focus on proteins that perform chemical reactions, enzymes.

**Classification of Enzyme Function.** ML models have emerged to predict the outcomes of enzymatic reactions [45, 46, 47] and for classification of enzyme function. Enzyme function is usually expressed using enzyme commission (EC) numbers, which is a hierarchical scheme for classifying enzyme function into classes (families) and consists of four levels of descriptions (Figure 1A). Some classification models are general protein function prediction models, which encompass all proteins, not just enzymes, such as ProtCNN/ENN [24] and ProteInfer [25]. Many models utilize representations from protein language models [48, 49, 50], and others incorporate protein structure as information, such as DeepFRI [23] and BioCLIP [51]. Methods related to supervised contrastive learning [52] have been particularly useful here, such as CLEAN, HiFi-NN, and Enzhier [22, 9, 53], likely by reducing imbalances in the number of sequences representing each EC number. Recently, enhanced approaches have enabled function prediction in ProtEx[26], PhiGnet [54], and others [55, 56, 57]. Other retrieval tools enable more sensitive detection of homologs such as DHR [58], ProtTrek [59], and using structure [17, 60, 61]. Finally, there are related models that predict substrates for enzymes [62, 63] and that aim to learn connections between chemical space and protein space [64].

**Large Language Models.** Recently, there has been an explosion in pretrained models in the biological and chemical domains, particularly large language models (LLMs) [65, 66]. For example, ChatGPT is capable of answering questions related to general scientific knowledge. These language models can be further finetuned for applications such as answering questions about protein sequences (including Pika [67], InstructProtein [68], InstructBioMol [69], ProteinGPT [70], and ProteinChat [71]) and for reaction synthesis planning (ChemCrow [72]), among others. LLMs present important benchmarks for enzyme functional classification and retrieval, given the widespread adoption of LLMs as science facilitators and their ease of use (in particular webserver-based approaches), compared to domain-specific methods [73].

**Multimodal Contrastive Learning.** Contrastive learning is an efficient and effective pretraining paradigm that aligns positive pairs and contrasts negative pairs simultaneously. The design of these pairs depends on the specific tasks, such as using data augmentations of the same image [74] or considering the topology and geometry of molecules [75]. More recently, contrastive learning has shown success in aligning the representation space of different biological and chemical modalities, *e.g.*, text and chemical structure alignment in MoleculeSTM [76], text and protein sequence alignment in ProteinDT [77] and ProteinCLIP [78], reaction structure and protein structure alignment in CLIPZyme [79], and protein sequence and structure alignment in BioCLIP [51] and with text in ProTrek [59], among others [80, 65]. Cross-modal alignment in the representation space has been shown to improve generalizability and improve performance on challenging tasks, such as out-of-distribution learning, zero-shot learning, and text-guided molecule design and optimization [77]. Consideration of multiple modalities may be especially important for the prediction of qualitative functions.

# 3 Overview of CARE

Though there are many studies using ML models to perform enzyme classification based on EC numbers (Figure 1A), there is no standardized benchmark to evaluate how well these models generalize to unseen protein sequences. To address this need, we present a benchmark suite for the classification and retrieval of enzymes (CARE, Figure 1B). CARE formalizes classification of an enzyme sequence by EC number as "Task 1" (Figure 1C). For this task, we design train-test splits of protein sequences to test out-of-domain generalizations that are relevant to real-world use cases. In addition, CARE addresses another key limitation of current studies: classification is limited to EC numbers, which is a closed vocabulary of functions (reactions), so existing models cannot generalize to unannotated reactions. Thus, we introduce an entirely new task, retrieval of an EC number given a reaction, which we call "Task 2" (Figure 1D). For this task, we design train-test splits to evaluate how well models can generalize to out-of-domain reactions, ensuring that the splits pose different levels of difficulty.

To streamline benchmarking, we curate a dataset of enzymes, reactions, and their associated EC numbers for CARE. At a high level, we build two datasets, one that links protein sequence to EC and one that links reaction to EC (Figure 1A). The former is processed from Swiss-Prot, the validated portion of UniProt [? ] and filtered to protein sequences between length 100 and 1024 with annotated

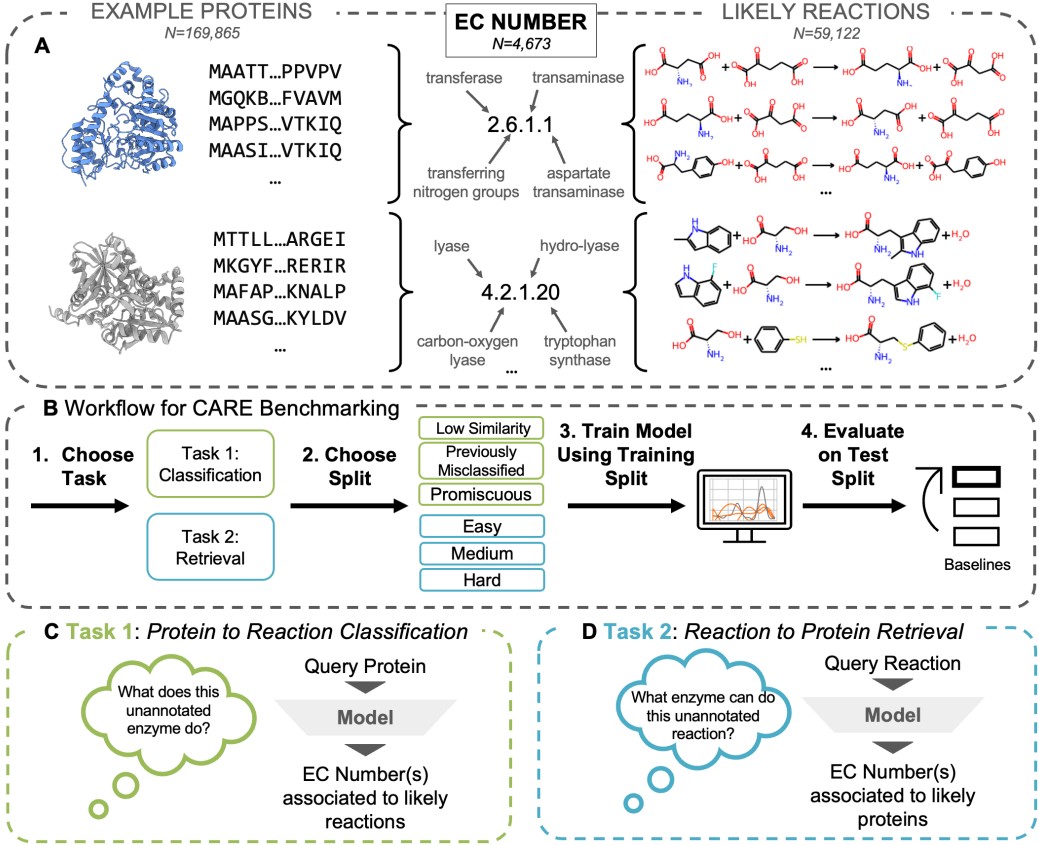

Figure 1: **Overview of CARE. (A)** Dataset format for CARE, showing examples of enzymes and their associated reactions. The EC number acts as a bridge between a protein sequence and the reactions it is likely to perform. The EC number is a hierarchical classification scheme for enzyme function with four levels of description, with increasing specificity from left to right. **(B)** General workflow for CARE benchmarking. **(C)** Task 1 is a real-world use case for enzyme classification based on a protein sequence. **(D)** Task 2 is a real-world use case for enzyme retrieval based on a reaction.

EC number(s). The latter is formed as a combination of EnzymeMap [37] and ECReact [36], where ECReact is only used to supplement EC numbers that are missing in EnzymeMap. Our workflow for generating the datasets used in this work is explained in detail in Appendix A.1 and shown visually in Appendix Figure A.1.

The overall workflow for benchmarking using CARE is shown in Figure 1B. For each task, domain-specific train-test splits are provided from the processed datasets. Model training can use any of the data in the train split, and each model is evaluated on the associated test split. In the rest of this study, we explain the specific design choices used to generate train-test splits and analyze benchmarking results of state-of-the-art methods on these splits. The curated datasets and splits used in CARE can be accessed at `https://github.com/jsunn-y/CARE/`.

## 4 Task 1: Enzyme Classification

Task 1, classification of an enzyme sequence, tests the ability of a model to extrapolate to unseen protein sequences. Task 1 is a fairly well studied task [20], but model evaluation has not been previously standardized as there are many factors to consider, such as the distribution of sequences and functions in the test sets. Task 1 applies to use cases where a scientist is given an unannotated enzyme sequence and seeks to understand the enzymatic function associated with that sequence (Figure 1C), for example for metagenomic analysis or finding new enzymes for retrobiosynthesis. With the emergence of conditional generative models for protein sequences, it is also important to

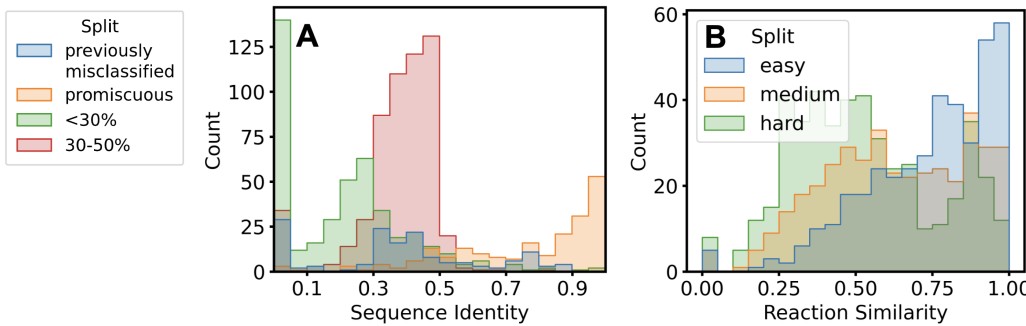

Figure 2: **Distribution of similarities between samples in each test set and the corresponding train set. (A)** Protein sequence identity (Task 1) was measured to the closest hit in the train set using BLASTp. Sequence identity can be thought of as normalized Levenshtein distance. **(B)** Reaction similarity (Task 2) was measured as cosine similarity with each reaction represented using DRFP [86]. Reaction similarity was measured from the test set to each train-set cluster center of reactions belonging to an EC number.

Table 1: **Summary of train-test splits used in Task 1.** For Task 1, certain protein sequences are held out. Train "samples" refers to the number of unique protein-EC data pairs, and test "samples" refers to the number of protein sequences. All splits for Task 1 share the same protein train set.

| Split name | Description | Train samples | Test samples |
|---|---|---|---|
| <30% Identity | Held-out sequences with approximately < 30% identity to the training set | 184,529 | 432 |
| 30-50% Identity | Held-out sequences with approximately 30-50% identity to the training set | 184,529 | 560 |
| Previously Misclassified (Price) | Previously misclassified enzyme sequences from Price et al. | 184,529 | 148 |
| Promiscuous | Held-out sequences with multiple EC numbers | 184,529 | 209 |

have high-throughput computational methods that can predict the function of generated sequences [81, 82, 83, 84, 85]. For this task, a query protein is passed through a trained model to predict an EC number. The EC number is then associated with likely reactions that the protein will be able to perform.

**Splits for Task 1.** Task 1 can be framed as evaluating how models generalize toward unseen sequences with different types of difficulty, visualized in Appendix Figure A.3A. The train-test splits for Task 1 are summarized in Table 1.

- *<30%* and *30-50% identity* splits: these two test splits contain sequences with sequence identities falling in the respective range, to sequences in the training set by using clustering (Section A.2). Sequence identity is related to the normalized Levenshtein distance between two sequences. Natural sequences with high (>40%) sequence identity at the protein sequence level are likely to share function [87]. It is expected that the lower the sequence identity, the more difficult it is to assign functional annotation.
- *previously misclassified (Price et al.)* [7] split: challenging to assign because some have low sequence identity to other proteins, and many may lie near "activity cliffs" (Figure A.3), a region where function can change sharply in sequence space.
- *promiscuous* split: in this study, we define promiscuous enzymes as those mapping to multiple EC numbers (thus lying on multiple activity peaks). These enzymes are particularly interesting for enzyme engineers, as new-to-nature activity can be found in between activity peaks [13].

For Task 1, the training split is constructed by holding out all of the pooled sequences in the test splits. The classification output vocabulary (EC numbers) is closed rather than open, so EC numbers in the

Table 2: **Performance of various methods on Task 1.** Performance is measured as k=1 classification accuracy (%). However, for the promiscuous split, we use k=(number of true ECs) for Random and CLEAN and report the average accuracy across all true EC numbers. For this split, we did not evaluate ChatGPT and Pika, as we prompted for only a single EC in the response. For BLAST and Foldseek, we use all EC numbers associated to the top single hit. More details are provided in Section A.4. CLEAN is a state-of-the-art method at the time of publication. Other methods such as ProteInfer, HiFi-NN, Enzhier, ProtEx, PhiGnet, etc. could also be benchmarked here.

| Split | Method | Level 4 Accuracy (X.X.X.X) | Level 3 Accuracy (X.X.X.-) | Level 2 Accuracy (X.X.-.-) | Level 1 Accuracy (X.-.-.-) |
|---|---|---|---|---|---|
| <30% Identity | Random | 0.0 | 1.2 | 3.2 | 19.4 |
| | BLASTp | 51.4 | 60.0 | 62.5 | 65.7 |
| | Foldseek | 54.9 | 68.3 | 72.9 | 80.6 |
| | ChatGPT | 0.0 | 0.0 | 1.6 | 28.9 |
| | Pika | 20.6 | 37.7 | 46.1 | 61.6 |
| | CLEAN | **55.1** | **68.8** | **74.8** | **84.5** |
| 30-50% identity | Random | 0.0 | 0.7 | 3.6 | 22.5 |
| | BLASTp | **81.1** | 87.9 | 90.7 | 92.3 |
| | Foldseek | 79.8 | 87.1 | 90.5 | 95.0 |
| | ChatGPT | 0.0 | 1.4 | 3.0 | 34.8 |
| | Pika | 37.7 | 50.2 | 60.0 | 73.8 |
| | CLEAN | 80.2 | **88.0** | **91.6** | **95.5** |
| Previously Misclassified (Price) | Random | 0.0 | 0.7 | 4.7 | 22.3 |
| | BLASTp | 35.1 | 70.9 | 78.4 | 78.4 |
| | Foldseek | **41.2** | **82.4** | **93.2** | **96.6** |
| | ChatGPT | 0.0 | 9.5 | 17.6 | 37.2 |
| | Pika | 4.1 | 50.7 | 64.9 | 82.4 |
| | CLEAN | 31.8 | 74.3 | 81.8 | 85.8 |
| Promiscuous | Random | 0.5 | 4.1 | 9.0 | 41.1 |
| | BLASTp | **93.7** | **94.8** | **95.2** | **95.9** |
| | Foldseek | 88.0 | 90.9 | 92.4 | 94.4 |
| | CLEAN | 69.4 | 77.9 | 81.5 | 87.0 |

test sets are present in the train sets. We verify that sequences in the test sets generally fall within the expected sequence identities, relative to the training set (Figure 2A). Notably, the *Price* test set has a wide distribution of sequence identities, while the *promiscuous* test set has high sequence similarity to the training set. The sequences in the test splits are distributed generally evenly across all different EC numbers (Figure A.4 in Appendix). More details can be found in Appendix A.2.

**Task 1 benchmarking results.** Benchmarking results for Task 1 are summarized in Table 2. We start with two methods as baselines, classification using a random order of EC numbers (Random), and Diamond BLAST at the protein sequence level, herein referred to as BLASTp, which is a workhorse bioinformatics tool that performs local-alignment to determine the most similar sequence(s) given a target query and a database [14]. Additionally, we include search by structural similarity using Foldseek [16].

While there exist many ML tools to directly perform EC number classification, CLEAN [22] and several others [53, 9, 26, 54, 57] are a few that seem to report the current state-of-the-art with comparable performance. Some of the latter methods were not yet publicly available at the time of this study, and other recent retrieval methods would also be promising approaches to test, but we opted to focus on benchmarking on CLEAN here. CLEAN generally performs the best for enzymes with low sequence identity to known sequences, but the BLASTp and Foldseek baselines perform similarly. ChatGPT was also tested, but it appeared to often hallucinate as it was forced to provide an answer, with results similar to the random baseline. Pika seems to bridge the gap between standard LLMs and enzyme classification, but its performance is not as good as standard tools like BLASTp [67].

Table 3: **Summary of train-test splits used in Task 2.** Certain reactions are held out, and "samples" refers to the number of reaction-EC pairs.

| Split name | Description | Train ECs | Train samples | Test samples |
|---|---|---|---|---|
| Easy | Certain reactions are held out, sampled uniformly across ECs, but no EC numbers are held out. The test set is the same as the holdout set. | 4,960 | 61,373 | 393 |
| Medium | All reactions corresponding to certain ECs are held out, at EC level 4 (X.X.X.X). Test set reactions are sampled uniformly across ECs from the holdout set. | 4,748 | 57,691 | 393 |
| Hard | All reactions corresponding to certain ECs are held out, at EC level 3 (X.X.X.-). Test set reactions are sampled uniformly across ECs from the holdout set. | 3,052 | 35,252 | 460 |

Overall, these results suggest that there is still room to improve classification of protein sequences with low sequence identity (*<30%*) and that lie near multiple activity peaks (*Price*). In fact, Foldseek has the best performance on the *Price* split, suggesting that structure can be conserved, even in scenarios where sequence is misleading. The *promiscuous* split is not easy for all methods; even though the test sequences have high sequence identity to the train set, CLEAN sometimes misses EC numbers. Interestingly, BLASTp and Foldseek perform close to the state-of-the-art, even when compared to more complex ML models. Future models could take advantage of this finding to augment training, as ProtEx does [26]. Additional details on the implementation of each method can be found in Appendix A.4.

## 5 Task 2: Enzyme Retrieval

Task 2, enzyme retrieval from a query reaction, tests the ability of a model to extrapolate to unseen reactions. Task 2 has not been formalized or explored in previous studies, but it is equally as important as Task 1, as it applies to a use case where a scientist or engineer is seeking to identify a previously characterized enzyme sequence that can perform a novel (unannotated) reaction (Figure 1D). Typical applications include: an environmental engineer looking for an enzyme to degrade a toxic pollutant [88], an enzyme engineer looking for an enzyme to catalyze a selective reaction for drug synthesis [89, 90], or a gene annotator identifying the gene for an "orphan" enzyme with known function but unknown sequence [91]. For Task 2, a query reaction is passed through a trained model to perform retrieval to an EC number and its associated proteins that are likely to be able to perform that reaction.

**Splits for Task 2.** Task 2 aims to evaluate how well a model generalizes to unseen reactions with different levels of difficulty. The train-test splits for Task 2 (*easy*, *medium*, and *hard*) are summarized in Table 3 and visualized in Appendix Figure A.3B. We equate greater difficulty with a more challenging train-test split; in a harder set, the test reactions are less similar to reactions in the corresponding train set. We decide similarity based on the amount of overlap in EC number (e.g., a reaction from 4.2.1.20 is considered more similar to another one in 4.2.1.20 than one from 4.2.1.1).

- *easy* split: EC numbers are randomly sampled at EC level 4 (X.X.X.X) and then randomly mapped to reactions, which are held out as the test set.

- *medium* split: the same reactions are used for testing as the easy set, but all other reaction-EC pairs which share the same EC level 4 (X.X.X.X) are held out from training.

- *hard* split: random EC numbers are sampled at EC level 3 (X.X.X.-) and all reactions under that EC3 are held out from training, while a subset of reactions from the held-out EC numbers are used for testing.

The sequences in the test splits are distributed generally evenly across different EC numbers (Appendix Figure A.4). From *easy* to *medium* to *hard*, the test set reactions also become more dissimilar to their respective training sets (Figure 2B). Reaction similarity was quantified by DRFP, which is a reaction representation that uses set differences between product and reactant fingerprints and has demonstrated solid performance without requiring model training [86]. Overall, Task 2 is more

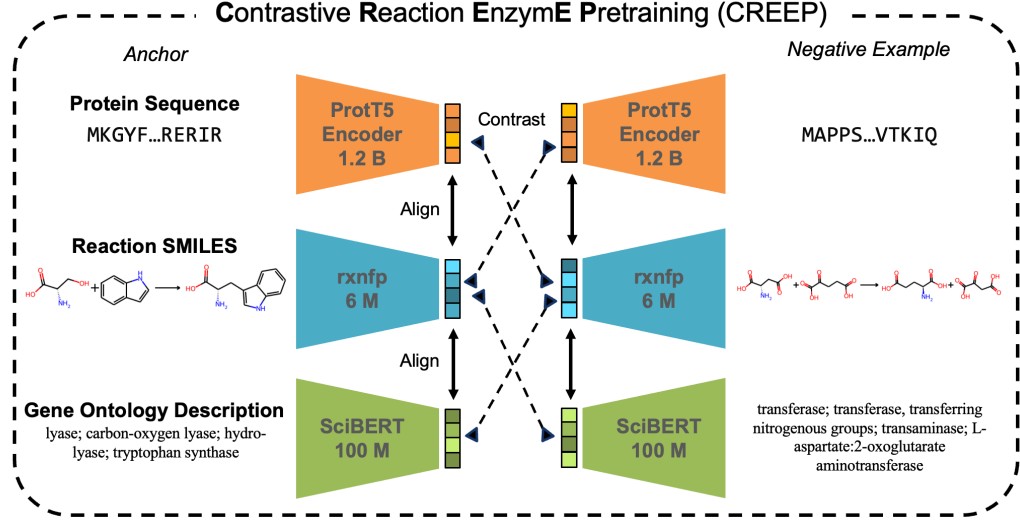

Figure 3: **Model architecture for CREEP.** CREEP aligns reaction and protein sequence using contrastive learning to perform downstream retrieval between domains. Optionally, CREEP training can be augmented by using textual description as a third modality to bridge the other two. Model sizes by number of parameters are also listed.

complex than Task 1, because unlike Task 1, entire EC numbers are held out from the training set (i.e., the classification output vocabulary is open rather than closed), so multiple modalities must be considered to link the unseen reactions to their respective EC numbers. More details on splitting can be found in Appendix A.2.

**CREEP baseline method.** There is a lack of models that have been tested for their ability to generalize beyond annotated reactions and none that have considered reaction, text and sequence together, so we develop a contrastive learning method for this task, called **C**ontrastive **R**eaction-**E**nzym**E** **P**retraining (CREEP) (Figure 3). Our approach is related to CLIPZyme [79], which uses contrastive alignment of reactions represented as 2D graphs and protein structures represented as 3D graphs. Somewhat differently, CREEP leverages finetuning of pretrained language models, rxnfp [92] and ProtT5 [50] to learn aligned representations of reactions and proteins, respectively. Rxnfp is a BERT-style [93] language model that was trained on reactions represented as SMILES/SMARTS strings and has demonstrated state-of-the-art performance on reaction type classification [92]. ProtT5 is a T5 protein language model that has been used for prediction of various protein properties and demonstrates similar performance to ESM [50, 49]. We used these language models for simplicity and their ease of finetuning. Uniquely, CREEP can learn on a third modality (CREEP w/ Text): textual description of the EC number based on gene ontology [94], which is encoded using SciBERT [95]. The use of text has recently been show to boost function prediction tasks [96]. More details on CREEP training can be found in Appendix A.3.

**Task 2 benchmarking results.** Benchmarking results for Task 2 are summarized in Table 4. We start with two methods as baselines: randomly guessing a ranking of EC numbers (Random) and finding the most similar reaction in the train set to the test query (Similarity Baseline). For the Similarity Baseline, we used DRFP to represent chemical reactions [86]. Details on the downstream multimodal retrieval process can be found in Appendix A.3 and Figure A.4. Methods like CREEP and CLIPZyme retrieve reference protein sequences based on a query reaction, and those proteins are mapped to a ranking of retrieved EC numbers; thus, they can generalize to EC numbers that are not linked to reactions in (i.e. missing from) the training set.

Overall, the Similarity Baseline is quite strong and performs much better than Random. ChatGPT was prompted with reactions written using compound IUPAC names rather than SMILES strings, both with (ChatGPT w/ Text) and without (ChatGPT) textual descriptions from gene ontology [94]. We only show ChatGPT performance for the easy split, as ChatGPT is not a good measure of generalization on the medium and hard splits. These both require "leaving out" entire EC numbers before training to ensure there is no data leakage.

Table 4: **Performance of various methods on Task 2.** Performance is measured as k=1 retrieval accuracy (%). *denotes that there may be data leakage causing performance to be inflated. Bolded accuracy is the best model.

| Split | Method | Level 4 Accuracy (X.X.X.X) | Level 3 Accuracy (X.X.X.-) | Level 2 Accuracy (X.X.-.-) | Level 1 Accuracy (X.-.-.-) |
|---|---|---|---|---|---|
| Easy | Random | 0.0 | 1.0 | 4.6 | 22.9 |
| | Similarity Baseline | 59.3 | 77.1 | 85.2 | 90.6 |
| | ChatGPT* | 4.8 | 22.6 | 43.5 | 71.0 |
| | ChatGPT (w/ Text)* | 13.7 | 56.7 | 81.4 | **98.5** |
| | CLIPZyme | 12.2 | 39.9 | 61.8 | 79.9 |
| | CREEP | 39.4 | 66.4 | 79.9 | 92.9 |
| | CREEP (w/ Text) | **60.3** | **89.3** | **93.9** | 96.7 |
| Medium | Random | 0.0 | 0.5 | 4.1 | 17.0 |
| | Similarity Baseline | 0.0 | 40.2 | 55.7 | 73.3 |
| | CLIPZyme | 2.0 | 26.0 | 46.6 | 69.0 |
| | CREEP | 4.1 | 44.3 | 63.1 | 86.5 |
| | CREEP (w/ Text) | **7.4** | **59.5** | **75.6** | **92.1** |
| Hard | Random | 0.0 | 0.9 | 1.5 | 18.3 |
| | Similarity Baseline | 0.0 | 0.0 | 13.5 | 42.0 |
| | CLIPZyme | 1.1 | 4.1 | 13.5 | 46.7 |
| | CREEP | 1.3 | 4.8 | 18.7 | **57.6** |
| | CREEP (w/ Text) | **1.3** | **9.8** | **22.2** | 57.2 |

On the more difficult test sets, CREEP offers an advantage compared to the Similarity Baseline, and particularly when combined with textual description: CREEP (w/ Text). This suggests that utilizing protein sequence information as a modality is useful. Still, performance on the harder splits is weak across the board, which suggests that there is significant room for future improvement, although retrieval of EC class may not be the ideal metric for out-of-distribution reaction classification. We anticipate that contrastive alignment with textual descriptions will play an increasingly important role in enzyme retrieval [78, 77, 65], and there is opportunity for better curation of these descriptions. Alternatively, CLIPZyme [79] represents reactions and proteins as graphs, but performance is lower, potentially due to some sample loss from some data being unable to be processed to graphs during training and inference. Here, performance could likely be improved by optimizing training hyperparameters. Additional details on the implementation of each method are in Appendix A.4, and additional benchmarking results are presented in Figure A.6.

## 6 Discussion

We made an important design choice in the CARE benchmarks: to perform classification and retrieval at the coarse-grained level of EC numbers, which could be a limitation. Enzymes can often perform many reactions, meaning it is an acceptable assumption that enzymes belonging to the same EC number will share the capacity to perform similar reactions, even if they are not directly annotated for all reactions. The ultimate task in this domain is to perform direct reaction to protein sequence retrieval and vice-versa (as in CLIPZyme [79] and Reactzyme [97]). However, currently the data for validation is limited, and we believe that many negative examples in those datasets may actually be feasible protein-reaction pairs. As experimentalists obtain higher resolution annotations of proteins and their associated reactions, this will become more realistic. Furthermore, our goal was to provide benchmarks for a broad range of functions, but model evaluation could also be performed on more specific classes of enzymes with direct protein-reaction pairs, as explored by EnzymeCAGE [98]. Other ways to hold out proteins with difficult-to-predict functions could also be explored. Some EC numbers are also incorrectly annotated, which are discussed in more detail here [62].

In the future, the proposed train-test splits could be refined for both tasks. For the Task 1 splits, there is some leakage in the sequence identity, with some sequences in test sets lying outside of the enforced sequence identity ranges, likely due to the different sequence similarity algorithms used

(MMseqs2 vs Diamond BLAST). MMseqs2 utilizes cascaded clustering which pre-filters based on initial clusters in the target set [99], while BLAST attempts to identify the closest sequence within the specified set based on local similarity. Another limitation is that while there are no duplicate reactions present in both the train and test sets, some of the test reactions are very similar to reactions in the training set despite having different ECs. It would be beneficial to do a more detailed analysis of reaction similarities and explore other representations of reactions to understand which reactions can be considered equivalent. Many EC classes also involve multi-complex enzymes; in other words, certain subunits of these enzyme are not actually performing catalytic activity. Future work could filter out some of the non catalytically active subunits or act to specifically predict the catalytic subunit, or the entire complex based on a reaction. Over time, the train-test splits should be updated as additional functional annotations are acquired and compiled in databases. For example, while over 36 million sequences in BRENDA/UniProt have EC numbers associated with them, these are often detected using homology based models, which may incorrectly assign EC numbers. A more detailed analysis of the specific failure modes of these ML models could be valuable future work [29].

We opted to assess performance using accuracy due to its simplicity and interpretability, but other retrieval and virtual screening metrics such as BEDROC and enrichment could also be explored. For the promiscuous enzymes with multiple EC numbers, we reported accuracy averaged across all of the true labels, but other classification metrics such as precision and recall should be considered in future work. The number of retrieved ECs could also be chosen using strategies like max separation and p-value as implemented in CLEAN [22].

There is also significant opportunity to use other modalities such as textual description and protein structure for more effective representations of enzymes and reactions, or to improve the text-based annotations. Gene context is also useful for annotating protein function and could be incorporated into future benchmarks [100]. The textual annotations used in this work are direct textual references to the EC class, meaning that the models in task 2 that incorporate text (e.g. CREEP), are effectively learning the relationship between the EC as a word and the EC as a number, in addition to the similarity of the reaction. Other tools such as Pika which use multiple types of textual description extend this to learn relations about the enzyme, and could be used in future iterations of CREEP. Future work will involve incorporating structure and graph based representations [101] into CREEP, similar to those used in CLIPZyme [79]. We also plan to do a more detailed analysis of the representations learned by CREEP. The addition of textual description in CREEP potentially introduces indirect data leakage between the train and test sets, so future iterations of CARE will need to consider this. Future work should also consider how to include protein function prediction models that go beyond enzymes–or models that would like to use additional data/modalities [59, 102]–into the CARE evaluation framework. LAB-Bench provides a framework for evaluating scientific reasoning using language, which will increasingly intersect with CARE [73]. A major bottleneck we encountered was that many models, including language models, were not available or difficult to use, limiting our ability to include them as benchmarks. There is also room to improve the prompt engineering of language models which could be further explored to enhance the performance of these models.

We finally note that methods used to retrieve dangerous proteins that could be used as bioweapons or to synthesize dangerous chemicals is a concern. The implications of this are discussed here [103].

## 7   Conclusion

Predicting the functions of enzymes is important for many applications ranging from gene annotation to enzyme engineering. While many models exist to classify enzyme function via EC numbers, there are no standardized benchmarks for evaluation of these models. Furthermore, no existing models have been tested for generalization beyond annotated reactions. To address this need, we introduce CARE, which is a benchmarking suite to formalize model evaluation for these two tasks. We also present CREEP, a model which uses multimodal contrastive learning and is one of the first models that can perform the latter task. We encourage developers to integrate their current and future methods or benchmarking results into the CARE Github repository (`https://github.com/jsunn-y/CARE/`). Overall, CARE is an important tool for encouraging progress in enzyme functional annotation. We believe that we are just seeing the beginning of the widespread adoption of multimodal models for protein functional prediction, and we expect that many researchers will find CARE useful for formulating and evaluating their models.

## Acknowledgments and Disclosure of Funding

This material is based upon work supported by the U.S. Department of Energy, Office of Science, Office of Basic Energy Sciences, under Award Number DE-SC0022218. This report was prepared as an account of work sponsored by an agency of the United States Government. Neither the United States Government nor any agency thereof, nor any of their employees, makes any warranty, express or implied, or assumes any legal liability or responsibility for the accuracy, completeness, or usefulness of any information, apparatus, product, or process disclosed, or represents that its use would not infringe privately owned rights. Reference herein to any specific commercial product, process, or service by trade name, trademark, manufacturer, or otherwise does not necessarily constitute or imply its endorsement, recommendation, or favoring by the United States Government or any agency thereof. The views and opinions of authors expressed herein do not necessarily state or reflect those of the United States Government or any agency thereof. J.Y. is partially supported by the National Science Foundation Graduate Research Fellowship. A.M. is supported by the Schmidt Science Fellows, in partnership with the Rhodes Trust. The authors thank Peter Mikhael and Itamar Chinn for helpful discussions on CLIPZyme implementation.

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

Figure A.1: Workflow used to process the datasets containing EC numbers, protein sequences, and reactions used in this study.

# A    Appendix

## A.1    Dataset Processing

Our data processing and splitting workflow is shown in Figure A.1. Protein sequence data, paired to EC number(s), were downloaded from UniProt, selecting only reviewed sequences in Swiss-Prot, resulting in a total of 571,282 sequences on the $13^{th}$ of May 2024. These were filtered to only retain sequences with an EC number that were not annotated as fragments. After, sequences were filtered to length between 100 and 1024, partial EC numbers (containing a dash) were removed, duplicate entries (based on EC and sequence) were dropped, and only EC numbers with associated reactions were kept, leaving 200,654 sequence-EC pairs. We call this the protein2EC dataset.

Reaction data were downloaded from ECReact and EnzymeMap. ECReact contains 62,222 entries of reaction in SMILES as SMARTS strings format coupled with EC number from a range of data sources. EnzymeMap had 349,458 entries containing sequence-reaction-EC triplets with the reaction in several formats, SMILES as SMARTS string, an atom-level mapping of bond formation and breaking (mapped reaction), and reactions written as text based on IUPAC compound names. Starting with EnzymeMap, protein sequence information was dropped (leaving reaction-EC pairs), and duplicate reaction-EC pairs were removed leaving 62,896 reaction-EC pairs. EC numbers that existed in ECReact but not in Enzyme map were added to EnzymeMap providing our reaction dataset of 72,490 reaction-EC pairs. We did this because EnzymeMap is a higher quality reaction dataset, with fewer incorrect entries. Note the ECReact only contains reactions formatted as SMILES/SMARTS, not as mapped reactions. Together, we call this the reaction2EC dataset.

Finally, we filtered the protein2EC and reaction2EC datasets, such that only EC numbers present in both datasets are retained, leaving 61,766 reaction-EC pairs and 185,995 sequence-EC pairs, with 4,960 unique EC numbers. Clustering using MMseqs2 [99] was performed at 30%, 50%, 70%, and 90% sequence identity and included in the protein2EC dataset for downstream use. The distributions of the curated datasets are shown in Figure A.2.

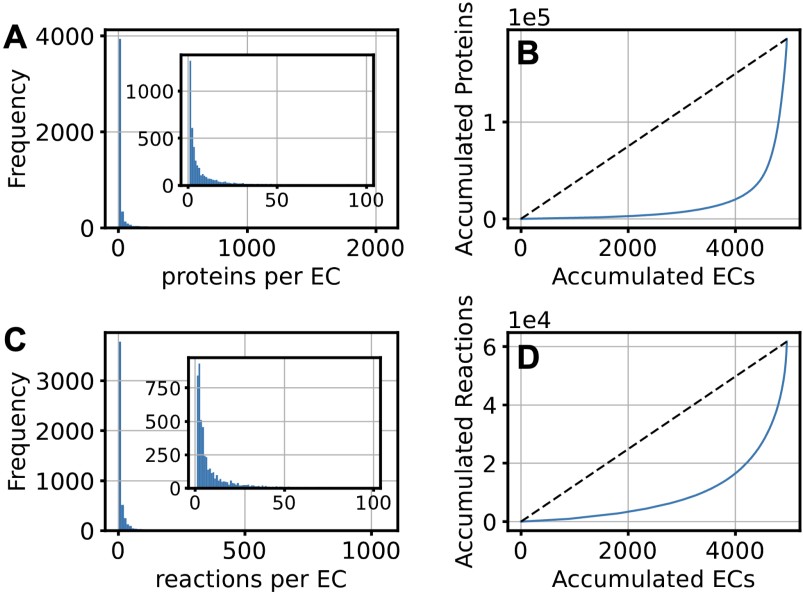

Figure A.2: Distribution of the protein2EC dataset, shown as (A) a histogram of proteins per EC and as (B) a cumulative distribution plot, analogous to a Lorenz curve. Distribution of the reaction2EC dataset, shown as (C) a histogram of reactions per EC and as (D) a cumulative distribution plot, analogous to a Lorenz curve. Both datasets are heavily skewed with most examples belonging to a few EC numbers, and many EC numbers only having one example. If samples were evenly distributed across ECs, they would follow the dashed line in (B) and (D).

## A.2    Dataset Splitting Details

First, we sought to create test sets capturing different types of out-of-domain generalization for Task 1, EC classification (visualized in Figure A.3A).

Using the results from clustering, we generated two test sets, with approximately <30% and 30-50% sequence identity, respectively, to sequences in the training set. We also only considered ECs at level 4 which had isolated clusters, in other words, an MMseqs2 cluster with only a single member. For every EC level 3 (X.X.X.-), we sampled up to three random ECs at level 4 (X.X.X.X), and then randomly selected a single isolated sequence from each class to add to the test set. This provided balanced test datasets across the functions at EC level 3. For this set, we also enforced that the sequences did not map to multiple EC numbers. Test sets were balanced across functions (e.g. EC number), rather than across their prevalence as protein sequences in nature, based on the idea that most synthesis planners and enzyme engineers would find more value in predictive ability across a broad range of functions.

Next, we looked for "promiscuous" enzymes, which in this work, we define as those annotated with multiple EC numbers. For our promiscuous test set, we only considered using promiscuous enzymes that had a combination of ECs that occurred at least twice. From this set we only took a single sample from each EC combination, and finally randomly selected a single entry from those which have ECs with high surprise levels (i.e. X.X.-.-, where variation occurs at the third EC level or higher). From the test sets, we did not include sequences that were labeled as belonging to part of a heterogeneous multi-complex protein.

Finally, we compiled a commonly misclassified test set from Price et al., filtered to sequences of length 100-1024, and dropped test sequences that are present in our protein2EC dataset. The pooled set of all proteins in the test sets were removed from our training data leaving 184,529 sequence-EC pairs for training. As a result of the holdout process, a few of the 4,960 EC numbers are missing from the train set, which should minimally affect performance on the test sets.

For Task 2, we sought to test reaction extrapolation, namely, retrieval of enzymes based on unseen reactions (visualized in Figure A.3B). For the easy and medium test sets, we randomly sampled an

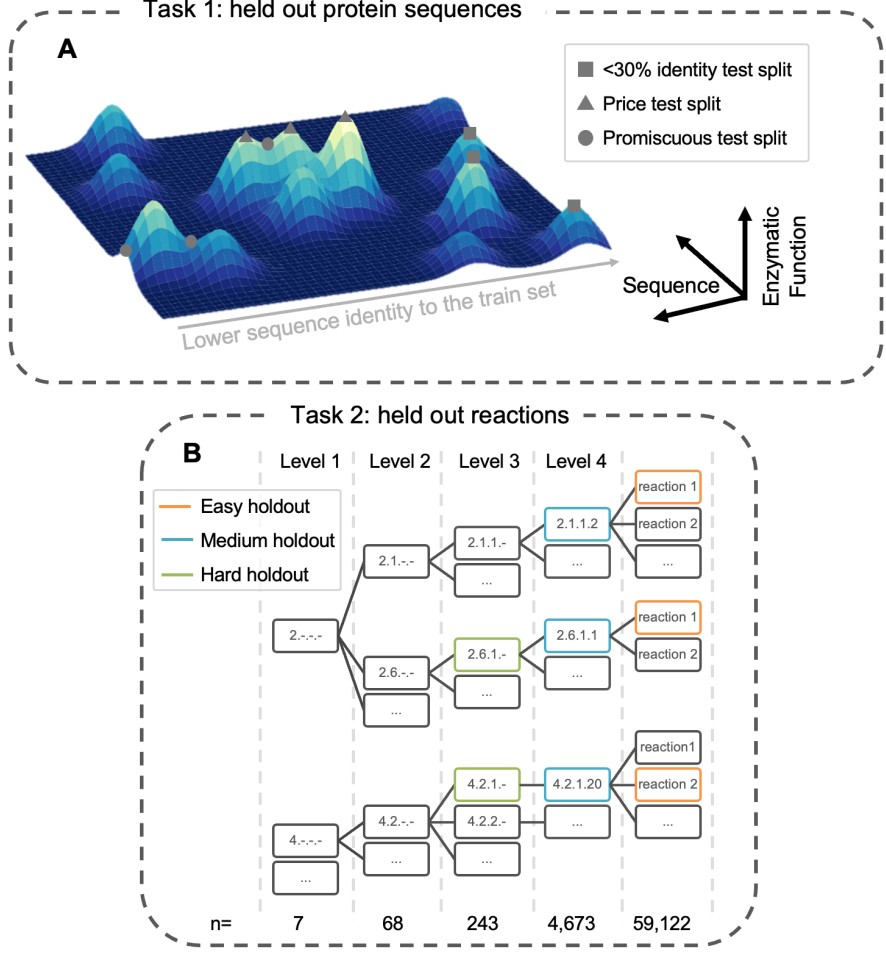

Figure A.3: (A) Visualization of the types of extrapolation found in the test splits for Task 1. (B) Visualization of how different subsets of reaction space are held out in the splits for Task 2.

EC level 4 EC (X.X.X.X) from every level three EC (X.X.X.-). From each of the held-out level 4 ECs, three random reactions were sampled to be in the test set. For the test set, we only considered (1) reactions that do not map to multiple EC numbers, (2) EC numbers with at least four reactions, and (3) reactions from EnzymeMap, which are higher fidelity and consequently have fewer misannotatations.

For the easy set, only the test set reactions were held out. For the medium set, all reactions under the corresponding level 4 ECs were held out. Note that the easy and medium test sets are exactly the same, but the easy set has fewer held out reactions compared to the medium set, thus their training sets are different.

The hard set was evenly balanced across a random sample from EC level 2 (X.X.-.-) to ensure there was an even distribution across level 3 (EC3, X.X.X.-), from which 53 random EC3s are sampled, with all reactions under the 53 EC3s held out. 460 reactions associated with the 53 ECs at level 3 were used as the test set, which are evenly spread across the ECs at level 3, with up to three reactions from each EC number at level 4. Note we ensure the hard test reactions are shared with the easy and medium ones, when possible, to correlate the performance of our test splits. It should be noted that the hard train-test split is different from the easy and medium splits. For Task 2, all of the protein-EC pairs can be used during training.

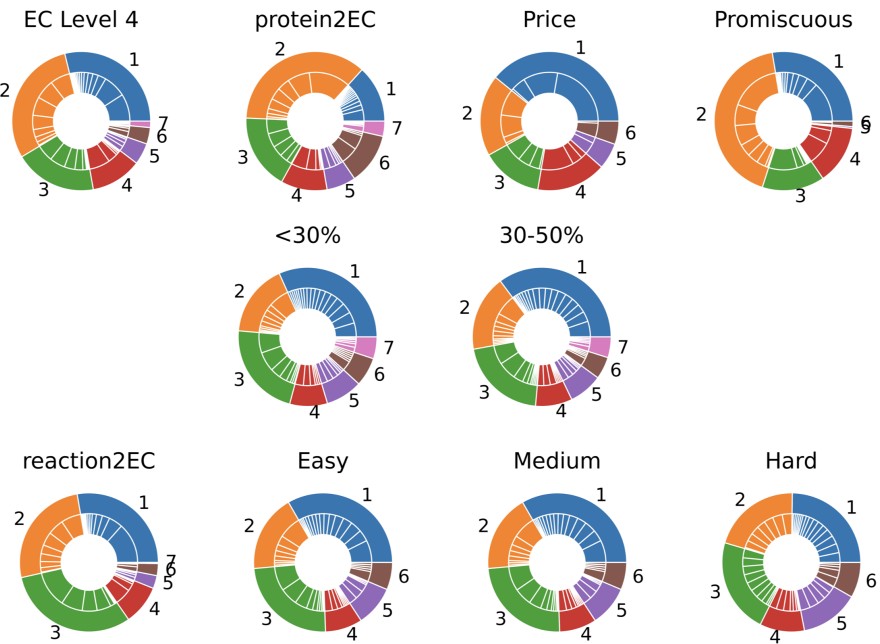

Figure A.4: Distribution of the EC numbers found in the test splits, and compared to a few reference datasets. Outer ring shows the level 1 category (X.-.-.-), and the inner ring shows the level 2 category (X.X.-.-). EC Level 4 refers the distribution of unique ECs at level 4 (X.X.X.X). protein2EC shows the distribution of all protein sequences in protein2EC across ECs, and reaction2EC shows the distribution of all reactions in reaction2EC across ECs.

## A.3 CREEP Model Details

All steps needed to reproduce CREEP training and downstream retrieval can be found at `https://github.com/jsunn-y/CARE/`.

During the pretraining stage, we follow the contrastive learning paradigm that maximizes mutual information between two views [104]. For each enzyme family (EC number) $x$, we extract three views: protein sequence $x_p$ (protein), reactions smiles $x_r$ (reaction), and textual description of EC number from gene ontology $x_t$ (text) [94]. In CREEP, the goal is to maximize the mutual information between all three modalities. For illustration purposes, we use the example of protein-reaction contrastive learning, but the derivations for reaction-text and text-protein contrastive learning follow a similar approach.

We follow the method proposed in GraphMVP by adopting EBM-NCE to estimate mutual information between our modalities [104]. In our case, EBM-NCE aligns the protein sequence and reaction SMILES pairs for the same enzyme family and contrasts the pairs for different enzyme families simultaneously. The objective function is

$$
\mathcal{L}_{\text{EBM-NCE}} = -\frac{1}{2}\Big(\mathbb{E}_{x_p,x_r}\big[\log\sigma(E(x_p,x_r))\big] + \mathbb{E}_{x_p,x'_r}\big[\log(1-\sigma(E(x_p,x'_r)))\big]\Big)
$$
$$
+ \mathbb{E}_{x_p,x_r}\big[\log\sigma(E(x_p,x_r))\big] + \mathbb{E}_{x'_p,x_r}\big[\log(1-\sigma(E(x'_p,x_r)))\big]\Big),
$$
(1)

where $x_p$ and $x_r$ form the (protein sequence, reaction SMILES) pair for each reaction, and $x_{p'}$ and $x_{r'}$ are the negative samples which are produced by randomly sampling from a Gaussian distribution, which we use as an approximation for the empirical data distribution. $E(\cdot)$ is the energy function with a flexible formulation, and we use the dot product as a metric for similarity within the latent space.

For our default CREEP model, we project representations to 256 dimensions in the shared latent space for all modalities. We train for 40 epochs, and in each epoch, we loop over each EC number and mine each batch such that all protein-reaction-text triplets come from different EC numbers. Note that we also use 50% sequence clustering to help select protein sequences with increased diversity.

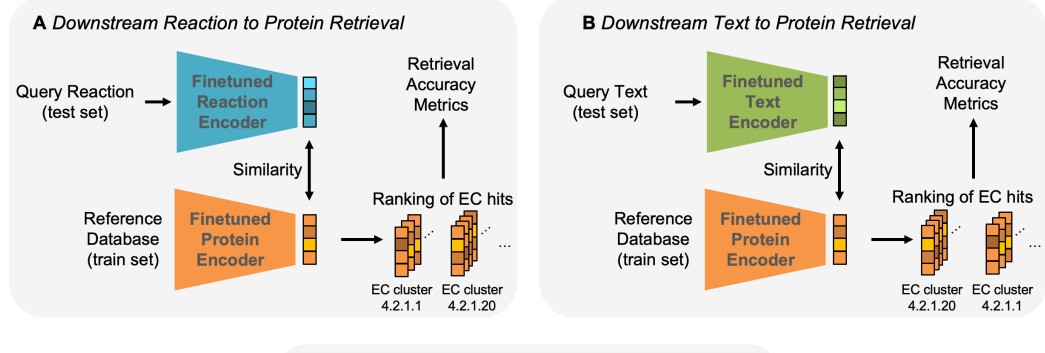

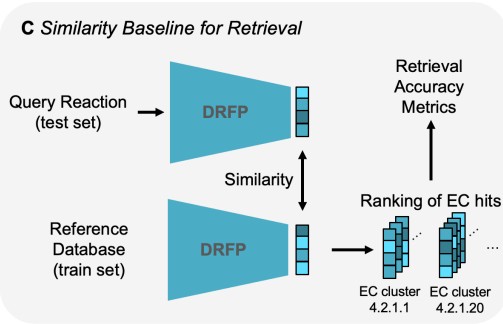

Figure A.5: Downstream retrieval using contrastively pretrained models. CREEP and CLIPZyme both utilize (A) reaction to protein retrieval. CREEP (w/ Text) can incorporate (B) text to protein retrieval, and consider the rankings from both (A) and (B). (C) The similarity baseline involves reaction to reaction retrieval using a fixed representation from DRFP [86].

Downstream retrieval using models such as CREEP and CLIPZyme is shown in Figure A.5. The standard workflow starts with a query reaction and ranks ECs by sorting based on distance from the reaction representation to the EC cluster centroids of protein sequence representations belonging to each EC number (Figure A.5A). For CREEP (w/ Text), a similar ranking can be determined by using text as a query (Figure A.5B), and the overall ranking can be calculated as an average of the two rankings. To reduce inference time, we used protein sequences clustered at 50% identity to calculate the protein representation centroids clustered by EC. Finally, the similarity baseline can be understood as sorting based on distance from a query reaction to reaction cluster centroids of reactions in the train set, clustered by EC (Figure A.5C).

## A.4 Implementation Details for Benchmarking Methods

All implementation details for methods used in the study can be found at `https://github.com/jsunn-y/CARE/`. We focus on evaluating the accuracy of retrieval for k=1. A level 1 accuracy means the top level EC class has to be correct, for level 2 accuracy, both level 1 and level 2 have to be correct, and so on. Accuracy is calculated as the number of true answers over the total number of examples. An invalid EC number from language models (ChatGPT and Pika) is considered incorrect on every level. For the non-promiscuous splits, the accuracy for each entry in the test set is 1 or 0. For the promiscuous split in Task 1, for each entry, we enumerate over each true EC to find the maximum accuracy within the pool of predicted ECs, and the overall accuracy for the entry is the average of the accuracies for each true EC.

**ChatGPT.** ChatGPT was used for both Task 1 and Task 2, using the API and gpt-4o-mini model. We performed minimal prompt engineering, with our initial prompt being the most standard initial question: "give me the top N EC numbers associated with this amino acid sequence:" for Task 1. See Table A.1 for example responses. The query was modified to: "You are protein engineer capable of predicting EC numbers from the sequence alone. You are also a skilled programmer and able to execute the code necessary to predict an EC number when you can't use reason alone. Given a protein sequence you are able to determine the most likely enzyme class for a sequence because you

are that skilled. You don't give up when faced with a sequence you don't know, you will use tools to resolve the most likely enzyme sequence. You only return enzyme commission numbers in a comma separated list, no other text is returned, you have failed if you do not return the EC numbers. You only return the most likely EC number." This resulted in a k=1 response, we found this more consistent then asking for a list of k results. Note that we do not do any filtering or data cleaning on the output of ChatGPT. Doing this could improve the accuracy, for example, in one case we identified in the hard set the output was EC4.6.1.18 which we scored as 0 despite the true EC being 4.6.1.2 for the reaction **UTP = 3',5'-cyclic UMP + diphosphate**. There are possibly other edge cases that we were unable to identify.

For Task 2, the query was maintained to be relatively consistent with the Task 1 query, and modified to: "Return the most likely EC number for this reaction: **naloxone + NAD(P)H = 6alpha-naloxol + NADP+**, which associates with the following text: **oxidoreductase; oxidoreductase, acting on CH-OH group of donors; oxidoreductase, acting on the CH-OH group of donors, NAD or NADP as acceptor; morphine 6-dehydrogenase.**", where a sample reaction has been bolded. Using the system prompt of: "You are protein engineer capable of predicting EC numbers from a combination of textual information and a reaction that corresponds to a specific protein. You are also a skilled programmer and able to execute the code necessary to predict an EC number when you can't use reason alone. Given a reaction and text information of an EC you are able to determine the most likely enzyme class for a reaction. You don't give up when faced with a reaction you don't know, you will use tools to resolve the most likely enzyme number. You only return enzyme commission numbers in a comma separated list, no other text is returned, you have failed if you do not return the EC numbers. You only return the most likely EC number." For Task 2 without the reaction text, we used the same prompt as above except with the reaction text components removed. Compared with the other tools, ChatGPT was provided with the reaction as a text string rather than a smiles string.

**BLASTp.** BLAST (Basic Alignment Search Tool) is an efficient algorithm for finding similar sequences in a reference database, to a query sequence [14]. We opted to use Diamond BLAST at the protein sequence level. For each test set, the training dataset was provided as the fasta file reference database, and the test set was provided as the query fasta. $k = 1$ hits were returned for each query, sorted based on similarity, using default parameters. The EC number classification was then inferred from the returned reference sequence(s).

**Foldseek.** Structures for entries in the test sets were obtained from the AF2 database. Missing structures were folded with ESMFold, but a few rare cases ( 5) could not be successfully folded. Structures for the Price dataset were obtained with ESMFold. Structures for sequences in the training set were obtained from the AF2 database. While a small fraction of structures in the training set could not be retrieved from the AlphaFold database, we decided to proceed without these. Structure search was performed with Foldseek with default parameters, and only the top hit was used to assign function.

**CLEAN.** CLEAN is a supervised contrastive model, which trains a classification head using embeddings from the ESM protein language model [48] by aligning embeddings from the same EC category and contrasting embeddings from different EC categories [22]. For Task 1, we retrained CLEAN using our training set with only one example from each cluster at 50% identity, following the instructions provided in their codebase. We did not perform any clustering before training. Because our training set size was similar in size to that used in the original model, we used the recommended 7000 epochs of training. We used the default script with triplet margin loss. After, we performed inference but added our own code into the original CLEAN code to output the classification results into a format compatible with our downstream analysis.

**Pika.** Pika is a finetuned LLM, which accepts protein sequence as an auxiliary input and can perform reasoning and protein function prediction [67]. We retrained Pika using the training set and evaluated it with the "qa" functionality by creating a new annotation set. The annotation set is an entry for each enzyme with the question: "What is the EC number of this protein?" and the answer of the EC number for that protein. For training, we use 70% of the training data for training, and 15% for testing and validation respectively (the "test" datasets are completely held out of this process). The model was trained for a maximum of 1000 epochs, with a maximum batch size of 100. We used the default models from Pika: esm2_t6_8M_UR50D as the protein model and gpt2 as the text model. All parameters were unchanged from the default apart from increasing the batch size and number of epochs, though hyperparameter tuning could improve the model. To evaluate the ability of Pika

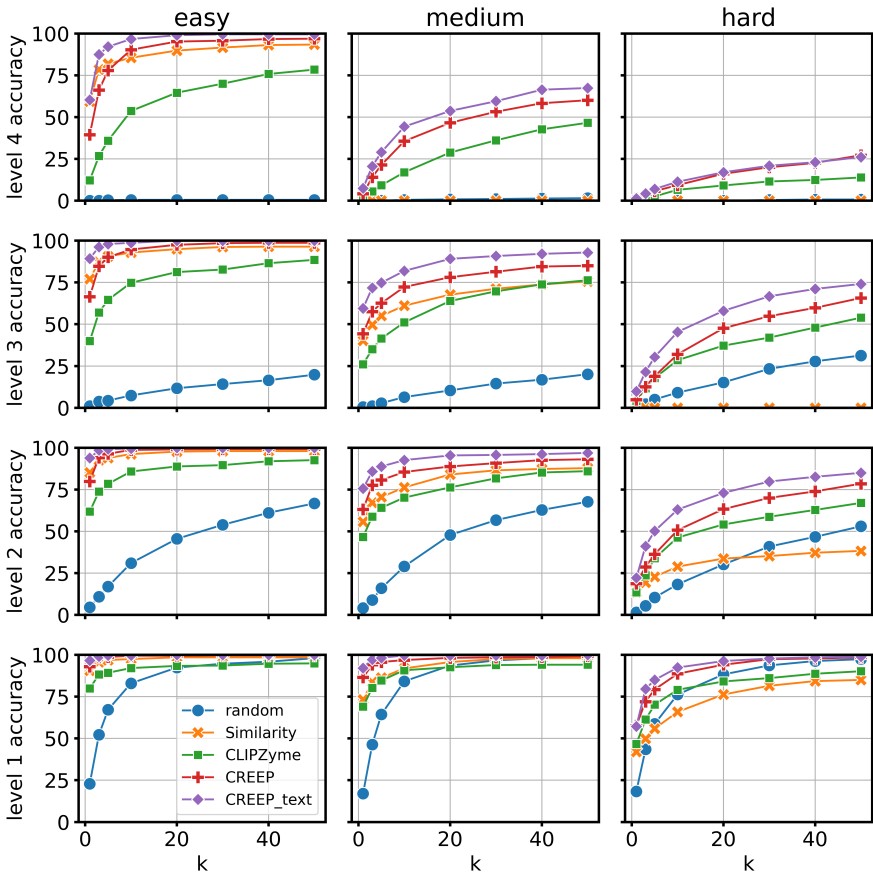

Figure A.6: Additional benchmarking results on Task 2, showing the increase in retrieval accuracy for $k = 1, 3, 5, 10, 20, 30, 40$.

to infer the EC numbers, we then query the trained model for each enzyme in the test set with the question: "What is the EC number of this protein?".

**CLIPZyme.**  CLIPZyme is a multimodal contrastive model, between reactions represented as 2D graphs and protein structures represented as 3D graphs [79]. CLIPZyme was trained on direct reaction-protein pairs from the EnzymeMap dataset [37]. In this study, we retrained CLIPZyme model using default parameters and the EGNN protein structure representation for retrieval on the Task 2 train-test splits. We modified the CLIPZyme code to loop over EC numbers and randomly sample pairs of protein sequences and reaction pairs associated with that EC number, much like CREEP, for 40 epochs. In this case, we only provided one protein sequence from each cluster at 50% identity, and we only trained with mapped reactions that had conserved atom numbers on both sides of the reaction, thus reducing the total number of available EC numbers. Furthermore, we dropped several protein sequences with structures that did not work with the EGNN encoder. In the future, the hyperparameters and training procedure could likely be optimized to yield improved performance.

**ChemCrow.**  ChemCrow is an LLM for chemistry problems, primarily synthesis planning [72]. It was unable to answer the results for Task 1, responding with the answers in Table A.1. For Task 2, we again tested a similar query to that input into ChatGPT, and received the responses in Table A.1. It should be noted, we use the public version so the private model may provide answers. Future work could involve building a biosynthesis-focused version of ChemCrow.

## A.5 Extended Benchmarking Results

Additional benchmarking results can be found at `https://github.com/jsunn-y/CARE/`.

Table A.1: Large language model responses to queries

| Tool | Query and Response |
|---|---|
| ChemCrow | *Return the EC number associated with this reaction: C=CCN1@]2..:* $\rightarrow$ Entering new RetryAgentExecutor chain... Thought: The question asks for the EC number associated with a given reaction. However, none of the provided tools can directly provide the EC number for a reaction. The EC number is typically associated with enzymes that catalyze reactions, and it's not something that can be derived from the reactants or products alone. Therefore, it's not possible to provide an answer using the available tools. Final Answer: I'm sorry, but with the tools available, I can't provide the EC number associated with the given reaction. The EC number is typically associated with enzymes that catalyze reactions, and it's not something that can be derived from the reactants or products alone.Thought: The question asks for the EC number associated with a given reaction. However, none of the provided tools can directly provide the EC number for a reaction. The EC number is typically associated with enzymes that catalyze reactions, and it's not something that can be derived from the reactants or products alone. Therefore, it's not possible to provide an answer using the available tools. Final Answer: I'm sorry, but with the tools available, I can't provide the EC number associated with the given reaction. The EC number is typically associated with enzymes that catalyze reactions, and it's not something that can be derived from the reactants or products alone. > Finished chain. "I'm sorry, but with the tools available, I can't provide the EC number associated with the given reaction. The EC number is typically associated with enzymes that catalyze reactions, and it's not something that can be derived from the reactants or products alone. |
| ChatGPT | *Examples of failed responses from ChatGPT:* **Example 1:**$\rightarrow$ Due to the limitations of this text-based platform, it's not possible to execute a program code here. In real conditions, I would use bioinformatics tools or a machine learning system to predict the Enzyme Commission (EC) numbers from the given protein sequence. However, in this case, I cannot complete this task. **Example 2**: $\rightarrow$ Due to the constraints of the current AI model, it's not able to computationally predict Enzyme Commission (EC) numbers from protein sequences directly. It's recommended to use specialized protein sequence analysis tools or machine learning algorithms designed for this purpose. For reliable and accurate results, databases like BRENDA, the Universal Protein Resource (UniProt), or tools like BLASTP, ECemble, DeepEC could be utilized. |

## A.6   License

The codebase is open source under the MIT license.

