# OpenReview forum: "CARE: a Benchmark Suite for the Classification and Retrieval of Enzymes"
_NeurIPS.cc/2024/Datasets_and_Benchmarks_Track — NeurIPS 2024 Track Datasets and Benchmarks Poster_

### Official Review · Reviewer_UrNk · 2024-07-20
**A dataset of Enzyme Commission numbers**

**Rating:** 5
**Confidence:** 5
**Clarity:** The paper is well written.

**Review:**

This paper presents a contribution to the field of computational enzyme function prediction. The work is well-motivated, clearly presented, and demonstrates a thorough understanding of the challenges in enzyme function prediction.

However, as a dataset paper, this work lacks novelty because the two datasets for task 1 and task2 have been curated in the previous work by CLEAN and Enzymemap respectively. The main contribution of this paper is clustering the sequences based on sequences similarity.

**Strengths:**

In this work, the authors have
1. Introduction of a novel task for enzyme retrieval based on reactions
2. Carefully designed train-test splits to evaluate out-of-distribution generalization
3. Built A new baseline model called CREEP for the retrieval task.

**Additional Feedback:**

Regarding to other LLMs for the protein retrieval task, one could get inspiration from this work: LAB-Bench (Laurent et al. 2024).

**Correctness:**

The evaluation methods and experiment design are appropriate and performed correctly.

**Documentation:**

It is unclear why the authors choose pretrained model ProtT5 over ESM for building CREEP.

**Ethics:**

I dont have ethical concerns.

**Limitations:**

1. It would be interesting to include recent models e.g. ProTrek (Su et al, 2024) in the model benchmark on the two tasks.
2. For the splitting, the authors used mmseq to cluster the proteins sequences. Previous work by FLOP (Groth et al. 2003) showed that it has limitations on clustering very diverse sequences.

**Opportunities For Improvement:**

Protein 3D structures of all these enzymes are available through alphafold but they havent being leveraged in this work.

**Relation To Prior Work:**

It is clearly discussed.

**Summary And Contributions:**

This paper introduces a benchmark suite designed to evaluate machine learning models for enzyme function prediction. The authors present two main tasks:
1. Classification of protein sequences by Enzyme Commission (EC) number
2. Retrieval of EC numbers given a chemical reaction

---

> ### Author Rebuttal · Authors · 2024-08-16
>
> **Review:**
> This paper presents a contribution to the field of computational enzyme function prediction. The work is well-motivated, clearly presented, and demonstrates a thorough understanding of the challenges in enzyme function prediction.
> * ***Our response:** We thank the reviewer for taking the time to understand our study. We have made various revisions to the manuscript to improve overall clarity, and we address specific suggestions and concerns point-by-point below.*
>
> However, as a dataset paper, this work lacks novelty because the two datasets for task 1 and task2 have been curated in the previous work by CLEAN and Enzymemap respectively. The main contribution of this paper is clustering the sequences based on sequences similarity.
> * ***Our response:** The reviewer raises valid points about the dataset itself not being new data. We believe that the contribution of our work lies in designing a new benchmark: standardized train-test splits that are framed under two tasks that encompass most real-world use cases. Secondly, linking Uniprot with EnzymeMap is a useful contribution, as current datasets do not bridge between most enzymes sequences and their respective reactions directly. Our benchmarking framework and the novel CREEP method will be useful for motivating future progress in this area. Based on this feedback, we are also adding a section to our github repo so that users can submit their own held out test sets for protein-reaction pairs which are not included in the current publicly available datasets, and we will update CARE accordingly over time.*
>
> **Opportunities For Improvement:**
> Protein 3D structures of all these enzymes are available through alphafold but they havent being leveraged in this work.
> * ***Our Response:** This is a good point. It is still an open question whether structural information can largely be captured by sequence, but it should be beneficial to use structure prediction and reaction graphs to directly map enzyme transition states, which is similar to the strategy employed by CLIPZyme (Mikhail et al., ICML 2024). This will be included in a future study building upon CREEP.  As for the tasks presented in CARE, researchers are welcome to use structure information in their models by linking PDB or AF database structures to the provided sequences.*
>
> **Limitations:**
> 1. It would be interesting to include recent models e.g. ProTrek (Su et al, 2024) in the model benchmark on the two tasks.
> 	* ***Our Response:** We thank the reviewer for this suggestion. We have revised our text to mention state-of-the-art methods released since submission such as ProTrek, PhiGNet, and others. Besides ProtTrek, there are also several methods released since submission which can retrieve homologs more sensitively than BLAST, such as DHR (Hong et al., Nat Biotech 2024) and FoldMason (Gilchrist et al., bioRxiv 2024). We are exploring these methods and will work to incorporate these baselines when possible. As growth in this field is rapid and it will be difficult to keep up with all new models coming out, we are also adding a section to our github repo so that users can more easily submit their benchmarking results on their new methods.*
> 2. For the splitting, the authors used mmseq to cluster the proteins sequences. Previous work by FLOP (Groth et al. 2003) showed that it has limitations on clustering very diverse sequences.
> 	* ***Our Response:** We thank the reviewer for this point. We chose to use MMseqs because it is the standard clustering tool. While it does have limitations, we verified with BLASTp that the generated test sets generally fall within their expected sequence identity thresholds (Figure 2A).*
>
> **Documentation:**
> It is unclear why the authors choose pretrained model ProtT5 over ESM for building CREEP.
> * ***Our Response:** This is a good question. In the original ProtT5 paper, transfer learning using the Prot-T5-XL encoder outperformed ESM1b on several tasks (Table 4 in Elnagger et al., IEEE 2021) A similarly sized model from ESM2, compared to Prot-T5-XL, performs similarly across several tasks (Table S3 in Lin et al., Science 2023). We have added a note of this in the revision. Thus we went with ProtT5 as we used it in a previous study involving contrastive learning (Liu et al., arXiv 2023). ESM2 would likely have yielded similar results, and we will explore this idea more in the future.*
>
> **Additional Feedback:**
> Regarding to other LLMs for the protein retrieval task, one could get inspiration from this work: LAB-Bench (Laurent et al. 2024).
> * ***Our Response:** We thank the reviewer for pointing us to this work, which is a comprehensive and relevant study. We are exploring whether these language models are capable of performing the tasks presented in CARE. Based on this sentence from LAB-Bench, there is still a lot of room for future improvement: “We also find, as expected, that models struggle to perform well on tasks requiring complicated manipulation of DNA and protein sequences, especially sub-sequences or long sequences.” We would also encourage future language model-based methods to submit their benchmarking results to our CARE github.*

---

> > ### Author Rebuttal · Authors · 2024-08-23
> >
> > We'd also like to note that we are currently adding novelty and protein engineering domain knowledge to construct better Task 1 splits:
> >
> > To address the reviewer's feedback, we are increasing the sizes of the 30%, 30-50% and promiscuous splits and adding in several subcategories that effect performance and are relevant for protein engineers to consider when choosing an annotation tool. These filters have not been considered before, but will make our splits more reflective of real world performance.
> > * 1) subunit performance, here the dataset will only contain subunits of larger enzyme structures, this can include regions without catalytic activity presenting a challenge to prediction.
> > * 2) fragments, performance on fragments presents information on how effective a tool is to noise in the dataset, it is common for reads to contain only partial genes so this is an interesting test case
> > * 3) proteins with active site annotations, this provides mechanistic information as to the activity of a protein.

---

### Official Review · Reviewer_QaD7 · 2024-07-24
**CARE review**

**Rating:** 7
**Confidence:** 4

**Review:**

Quality
- “validated portion of UniProt [23] and filtered to protein sequences between length 100 and 650 with annotated EC number(s).” — why did you remove proteins with >650? Typically plms remove >1024. What percentage of the data does this remove?
- The number of test samples is quite small in Table 1 (<1%), why is this the case?
- Should consider including naive baseline to always predict EC label with highest frequency in train to give signal of class balances
- What is the motivation for k=5 for promiscuous test set? if I remember correctly from the CLEAN paper, most multi-label proteins have 2 EC numbers (?), is it different in this dataset? For blast, do top 5 hits return 5 EC numbers or can they potentially map to the same EC number? Also, why do you choose this setup as opposed to not constraining the number of predicted EC labels (like in CLEAN) and then measuring F1 score?

Clarity
- When creating the test sets, should link to relevant clustering sections in appendix, since it wasn’t clear from the text how the similarity based test sets were initially generated (was only clear after I looked at appendix, but some explanation should be included in the main text)
- Line 120 “ECReact is only used to supplement EC numbers that are missing in EnzymeMap”: why do you only use ECReact to supplement and not take e.g., the set of their union? Is ECReact lower quality?
- What is the checkpoint date of UniProt that was used to generate the training/test sets?
- Can samples in training set have multiple EC numbers for task 1?
- I’m a bit confused about the setup for Task 2 — if you are removing entire EC numbers from the training set in e.g., the medium and hard splits, how are you able to predict those EC numbers for the test set?
- Line 48 + 220: authors are claiming CREEP is initial baseline but difference to CLIPzyme isn’t here, which seems to do the same thing. Can authors further clarify this?
Originality
- Authors introduce a new task for the community which is meaningful (given a new chemical reaction, what enzyme would catalyze it)

Significance
- There are an increasing number of papers trying to predict EC but no standard benchmarks (usually people have to format databases themselves) so this would be beneficial for the community

**Strengths:**

- Overall I think it was a very well-written paper and well motivated.
- I like the characterization of reaction similarity using DRFP in figure 2.

**Additional Feedback:**

- Would be interesting to compare to some other commonly used pLMs like ESM for Task 1 using nearest neighbour retrieval

**Clarity:**

Overall, the paper is well written and the appendix provides further detailed analyses.

**Correctness:**

Apart from my questions about the promiscuous test set and Task 2, the paper and dataset construction seem sound.

**Documentation:**

Following instructions from the git repo I was able to download the dataset without problems. The git repo seems to provide sufficient detail for rerunning experiments and authors provide Jupiter notebooks to demonstrate how to load things.

**Ethics:**

No ethical concerns.

**Limitations:**

Authors do not address negative societal impact of their work (they write it was described elsewhere but don’t point to another location). Please provide a link to this other location or describe in the current work.

**Opportunities For Improvement:**

- Unclear why test sets are so small in comparison to training for Task 1 and also why so many sequences were filtered out (Swiss-Prot normally contains 500k sequences, but this dataset contains 170000k, would appreciate some clarification on this)
- Task 2 — confused about how you can predict EC degree 4 on your test set with good accuracy if you don’t have them in your training set (and im assuming retrieval database) — can you please provide some clarification on this?

**Relation To Prior Work:**

The paper clearly discusses prior work well.

**Summary And Contributions:**

Authors introduce CARE, a dataset for (1) predicting enzyme commission numbers from a protein sequences, and (2) predicting enzyme commission numbers from a chemical reaction. Authors also introduce CREEP, a contrastive learning framework for solving Task 2 and show that it beats existing baselines with no data leakage.

---

> ### Author Rebuttal · Authors · 2024-08-16
>
> Authors introduce CARE, a dataset for (1) predicting enzyme commission numbers from a protein sequences, and (2) predicting enzyme commission numbers from a chemical reaction. Authors also introduce CREEP, a contrastive learning framework for solving Task 2 and show that it beats existing baselines with no data leakage.
> * ***Our response:** We thank the reviewer for their support of our study and detailed feedback. We have made various revisions to the manuscript to improve overall clarity, and we address specific suggestions and concerns point-by-point below.*
> **Quality**
> - “validated portion of UniProt [23] and filtered to protein sequences between length 100 and 650 with annotated EC number(s).” — why did you remove proteins with >650? Typically plms remove >1024. What percentage of the data does this remove?
> 	- ***Our response:** We filtered out long sequences to avoid complicated large enzymes such as gene-editing enzymes; many of these functions do not have associated reactions in EnzymeMap. In total, the sequence-based filtering removed about 10% of the training data. We aim to include these longer sequences in future updates to CARE as better reaction annotations become available.*
> - The number of test samples is quite small in Table 1 (<1%), why is this the case?
> 	- ***Our response:** We considered generating larger test sets to reduce statistical variation in performance, but we settled on these small test sets so that we can hold them out collectively and maintain a single train set without removing too much data from the train set. The goal is to make benchmarking more convenient for future users (as only one model will need to be trained for task 1). The test sets are distributed across functional space, so we believe that they are sufficiently diverse.
> - Should consider including naive baseline to always predict EC label with highest frequency in train to give signal of class balances
> 	- ***Our response:** Thank you for this suggestion. Our random baseline should capture this effect as we randomly sample from the EC numbers present in the training dataset. We would also note that the test sets are scattered evenly across the distribution of EC numbers, not the distribution of sequences, so our random baseline generally matches this skew.*
> - What is the motivation for k=5 for promiscuous test set? if I remember correctly from the CLEAN paper, most multi-label proteins have 2 EC numbers (?), is it different in this dataset? For blast, do top 5 hits return 5 EC numbers or can they potentially map to the same EC number? Also, why do you choose this setup as opposed to not constraining the number of predicted EC labels (like in CLEAN) and then measuring F1 score?
> 	- ***Our response:** Thank you for these questions. We chose k=5 because we specifically chose sequences for the promiscuous test set with as many EC numbers as possible, and thus many of these sequences have more than 2 EC numbers. For BLAST, it is correct that it is not the most apples to apples comparison because the top hits can potentially map to the same EC number (but each hit can have multiple EC numbers). We reported accuracy, as it is more interpretable, but this is highly correlated to the F1 score. We also did this because choosing the top n EC labels adds a level of complexity which is not relevant to many real world applications where promiscuity is useful. For example, if an enzyme engineer is looking for an enzyme with a target function, it is more important to evaluate whether the function is captured within some k hits. It matters less if the retrieval method mistakenly classifies sequences as having certain other functions, as long as it can identify the correct ones. Furthermore, certain methods such as BLAST are not directly able to learn how many EC numbers to retrieve. Still, we will add F1 score calculations into the revision for applicable methods (namely CLEAN) on the promiscuous test split.*

---

> > ### Author Rebuttal · Authors · 2024-08-16
> >
> > **Clarity**
> > - When creating the test sets, should link to relevant clustering sections in appendix, since it wasn’t clear from the text how the similarity based test sets were initially generated (was only clear after I looked at appendix, but some explanation should be included in the main text)
> > 	- ***Our response:** We thank the reviewer for this suggestion and have added a revision to the main text.*
> > - Line 120 “ECReact is only used to supplement EC numbers that are missing in EnzymeMap”: why do you only use ECReact to supplement and not take e.g., the set of their union? Is ECReact lower quality?
> > 	- ***Our response:** This is correct. ECReact is believed to be lower quality than EnzymeMap because it has not been atom-level mapped. The creators of EnzymeMap found that more of the reactions in ECReact are misannotated.*
> > - What is the checkpoint date of UniProt that was used to generate the training/test sets?
> > 	-  ***Our response:** The checkpoint date is May 13, 2024 and is provided in the Appendix.*
> > - Can samples in training set have multiple EC numbers for task 1?
> > 	- ***Our response:** Yes, several thousand sequences in the training set have multiple EC numbers.*
> > - I’m a bit confused about the setup for Task 2 — if you are removing entire EC numbers from the training set in e.g., the medium and hard splits, how are you able to predict those EC numbers for the test set?
> > 	- ***Our response:** This is a good question, and Figure A.5 provides a visualization of the retrieval process. For Task 2, while entire EC numbers and their associated reactions are held out from the training set, the proteins associated to these EC numbers are not. During the downstream retrieval step, the trained model can be used to retrieve from a queried reaction any protein, which includes those annotated with any left out EC numbers - thus allowing a model to extrapolate to the correct EC number. We have added a sentence to the main text to clarify this.*
> > - Line 48 + 220: authors are claiming CREEP is initial baseline but difference to CLIPzyme isn’t here, which seems to do the same thing. Can authors further clarify this?  Originality
> > 	- ***Our response:** The reviewer is correct in pointing out that CREEP and CLIPZyme are complementary approaches developed concurrently. The evaluation framework for CLIPZyme is somewhat different, and we believe that our evaluation framework in CARE is more well suited to real-world enzyme function prediction tasks. For example, the CLIPZyme dataset is very conservative about which protein-reaction pairs are valid, but due to the promiscuity of enzymes and a lack of all annotations, many correct protein-reaction pairs are missing. Furthermore, our approach leverages pretrained language models which may offer knowledge from unlabelled data. Our future studies will include combining the ideas used in CREEP and CLIPZyme to further improve performance.*
> >
> > **Opportunities For Improvement:**
> > - Unclear why test sets are so small in comparison to training for Task 1 and also why so many sequences were filtered out (Swiss-Prot normally contains 500k sequences, but this dataset contains 170000k, would appreciate some clarification on this)
> > 	- ***Our response:** There are about 220k enzymes in Swiss-Prot with EC numbers out of 500k total proteins, and we further filtered our dataset based on sequence length cutoffs and those that were associated with annotated reactions in EnzymeMap and ECReact (so 170K represents 77% of the original 220K in Swiss-Prot with EC numbers).  The exact process is  provided in our github and described in the appendix. We considered generating larger test sets to reduce statistical variation in performance, but we settled on these small test sets so that we can hold them out collectively and maintain a single train set without removing too much data from the train set. The goal is to make benchmarking more convenient for future users (as only one model will need to be trained). The test sets are distributed across functional space, so we think that they are sufficiently diverse.*
> > - Task 2 — confused about how you can predict EC degree 4 on your test set with good accuracy if you don’t have them in your training set (and im assuming retrieval database) — can you please provide some clarification on this?
> > 	- ***Our response:** Thank you for this question. Performance is only high (60%) for the easy split on task 2, because in this split there are not any EC numbers to degree 4 which are entirely head out. Performance drops significantly to 3.4% and 5.1% for CREEP when moving to the medium test split on task 2 because entire EC numbers are held out at level 4. There is a lot of room for improvement here, and we hope that researchers will use CARE to develop methods for this task.
> >
> > **Limitations:**
> > Authors do not address negative societal impact of their work (they write it was described elsewhere but don’t point to another location). Please provide a link to this other location or describe in the current work.
> > * ***Our response:** This is a good point, and we have included a stronger description of the ethical dangers in our revision, with citations.*
> >
> > **Additional Feedback:**
> > - Would be interesting to compare to some other commonly used pLMs like ESM for Task 1 using nearest neighbour retrieval
> > 	- ***Our Response:** Thank you for this suggestion. We note that CLEAN is one of the benchmarks we included. CLEAN effectively uses ESM embeddings combined with supervised contrastive learning and nearest neighbor retrieval, and it outperforms raw nearest neighbor retrieval from ESM.*

---

> > > ### Comment · Reviewer_QaD7 · 2024-08-30
> > >
> > > Thank you to the authors for addressing my concerns.
> > >
> > > >We filtered out long sequences to avoid complicated large enzymes such as gene-editing enzymes
> > >
> > > How did you decide on the number 650 in the first place? Also by applying this more strict filter, are you missing out on any larger enzymes that are biologically interesting/relevant?
> > >
> > > It might still be nice to include a widely-used PLMs (like ProtT5 if not ESM) as a comparison for Table 2 since these embeddings are widely used, but I will leave that to the authors' discretion. In general I am pleased with the responses and have increased my score.

---

> > > > ### Author Rebuttal · Authors · 2024-08-30
> > > >
> > > > Thanks for these questions and for your additional feedback. To be honest, 650 was relatively arbitrary and we should probably increase the sequence length limit to 1024. We will explore this in the revision.

---

### Official Review · Reviewer_hc5Q · 2024-07-24
**An ambitious and timely benchmarking suite for enzyme discovery and design**

**Rating:** 9
**Confidence:** 3
**Correctness:** Yes
**Clarity:** Yes

**Review:**

The quality of the work is outstanding. The paper is well reasoned and presented. Figures are well done and make the analysis pipelines very interpretable. The main originality in the work is Task 2, which provides a new benchmarking tool for retrieving enzymes that perform a particular reaction.

pros:
+well documented code base
+figures are well done
+benchmarking is compared not just to SOTA tools, but also to popular general use LLMs (chatGPT)

cons:
-new model CREEP does not inculde structural data, which is really key to protein function
-how currently useful are these more advanced models given that tools like BLASTp perform close to SOTA? Also, the success in the hardest Task 2 benchmark is modest at best.

**Strengths:**

The major strengths of this submission are the well designed benchmarking pipelines, and robust benchmark design using varying splits and task difficulty levels. This area of research is a really important particularly today and in the coming years as protein design tools mature, similar benchmarking methods and tools like CREEP could become useful there. This contribution should make this area of research more broadly accessible to the research community, and more fair in the comparisons.

Could there be social or ethical considerations if the retrieval tool was used to find potentially harmful enzymes? e.g. ricin-like reactions?

**Additional Feedback:**

see comments above, and also a wider discussion speculating what new types of data is needed in the field to push the Task 2 forward.

**Documentation:**

Yes code base was checked and appears adequate.

**Ethics:**

Minor concerns could be discussed such as how such tools could be used for finding potentially dangerous enzymes.

**Limitations:**

The lengthy discussion of limitations is appreciated.

**Opportunities For Improvement:**

The potential for data leak in some of the benchmarked tools is major problem that obscures interpretation. Is there anyway the authors can address this?  Structure determines function of a protein, so inclusion of explicit structure in the CREEP method is likely to improve it.

**Relation To Prior Work:**

Yes

**Summary And Contributions:**

This work provides a robust set of benchmarks in the area of enzyme classification -- that is 1) the ability to predict an enzyme's function given its sequence, and 2) the ability to retrieve sequences that might perform a given reaction. The latter is novel and particularly relevant given recent advancements in both metagenomic data, and the ability to design new proteins de novo. Benchmarks are provided on existing SOTA models that have been previously, and also on nascent LLMs such as ChatGPT, providing interesting out of the box comparisons. While the results are robust, it is clear these tasks remain challenging, especially task 2.

---

> ### Author Rebuttal · Authors · 2024-08-16
>
> This work provides a robust set of benchmarks in the area of enzyme classification -- that is 1) the ability to predict an enzyme's function given its sequence, and 2) the ability to retrieve sequences that might perform a given reaction. The latter is novel and particularly relevant given recent advancements in both metagenomic data, and the ability to design new proteins de novo. Benchmarks are provided on existing SOTA models that have been previously, and also on nascent LLMs such as ChatGPT, providing interesting out of the box comparisons. While the results are robust, it is clear these tasks remain challenging, especially task 2.
>
> **Review:**
> The quality of the work is outstanding. The paper is well reasoned and presented. Figures are well done and make the analysis pipelines very interpretable. The main originality in the work is Task 2, which provides a new benchmarking tool for retrieving enzymes that perform a particular reaction.
> pros: +well documented code base +figures are well done +benchmarking is compared not just to SOTA tools, but also to popular general use LLMs (chatGPT)
> * ***Our response:** We thank the reviewer for their enthusiastic support of our study. We have made various revisions to the manuscript to improve overall clarity, and we address specific recommendations point-by-point below.*
>
> cons:
> * new model CREEP does not include structural data, which is really key to protein function
> 	* ***Our response:** This is a good suggestion. It is still an open question whether structural information can be largely captured by sequence; however, CLIPZyme (Mikhael et al., ICML 2024) shows that using structure prediction and reaction graphs to encode enzyme transition states can improve model performance. We will likely explore similar approaches in a future study that extends CREEP.*
> * how currently useful are these more advanced models given that tools like BLASTp perform close to SOTA? Also, the success in the hardest Task 2 benchmark is modest at best.
> 	* ***Our response:** CLEAN is generally more useful than BLAST and is more consistent at classification of EC numbers as coarser levels (such as EC level 3 X.X.X.-). It is correct to point out that there is clearly a lot of room for improvement in Task 2, and we hope that researchers will be able to use CARE to make progress in this area!*
>
> Could there be social or ethical considerations if the retrieval tool was used to find potentially harmful enzymes? e.g. ricin-like reactions?
> * ***Our response:** This is a good point and we have included a stronger description of the ethical dangers in our revision.*
>
> The potential for data leak in some of the benchmarked tools is major problem that obscures interpretation. Is there anyway the authors can address this?
> * ***Our response:** We thank the reviewer for this suggestion. Since submission, we are working on retraining existing models such as ProteInfer and CLIPZyme to provide a more interpretable comparison. These results will be added in the revision or an updated release to the package. We have also provided a more thorough description of existing state-of-the-art methods which were not publicly available at the time of submission such as HiFi-NN, ProtEx, PhiGnet, and others. Finally, our goal in the development of this benchmark was that by making the datasets used for training and testing easily accessible, new models with benchmarking results can be directly added to the github by developers without data leakage in the future.*
>
> Structure determines function of a protein, so inclusion of explicit structure in the CREEP method is likely to improve it.
> * ***Our response:** Thank you for this suggestion. Our future work will improve CREEP by adopting principles from CLIPZyme such as the incorporation of protein structural information and reaction graphs.*

---

> > ### Comment · Reviewer_hc5Q · 2024-08-23
> >
> > The authors have addressed my minor concerns, and my original very supportive score stands.

---

### Decision · Program_Chairs · 2024-09-26

**Decision:**

Accept (Poster)

**Comment:**

Summary:
This paper presents a benchmarking suite focusing on enzyme function prediction with two main tasks: (1) Predicting enzyme commission (EC) numbers from protein sequences; and (2) Predicting EC numbers from chemical reactions, including retrieval of sequences for specific reactions. The work is valuable for its novel approach to benchmarking enzyme retrieval and attempts to address challenging computational biology questions. It has garnered positive reviews for its detailed benchmarks and the introduction of the CREEP model. However, some key limitations and areas of improvement have been identified which, if addressed, could significantly strengthen the paper.

Strengths:
The benchmarking tasks are well-designed and pose significant value to the enzyme discovery and design community.
Introduction of a novel retrieval task and careful design considerations of train-test splits.
Positive contributions through the integration of new benchmarking tools compared to state-of-the-art (SOTA) models, including LLMs such as ChatGPT.
Clear presentation with well-reasoned methodology and support from robust figures and documentation.

Areas for Improvement and Suggested Revisions:

Inclusion of Structural Data in CREEP:
The incorporation of 3D structure data (e.g., from AlphaFold predictions) could improve the accuracy and relevance of the CREEP model. Future work should consider integrating such data.

Clarification of Dataset Choices:
The rationale for excluding longer sequences (>650 residues) should be clarified, along with consideration of their inclusion if they are biologically relevant.
Explanation for the relatively small test set size compared to training data. Consider expanding test sets for robustness or provide a stronger justification for current choices.
Baseline and Comparison Improvements:

Include naive baseline comparisons to highlight class imbalances explicitly.
Consider comparing to additional SOTA models like ProtT5 or ESM embeddings.
Data Leakage and Ethical Implications:
Address potential data leakage issues more explicitly in the methods and results. Ensure detailed documentation is available to guide users in avoiding data leakage.
Expand on the ethical discussion regarding the potential misuse of such tools, specifically in developing harmful enzymes.

Technical Clarifications:
Provide additional clarity on how task 2 predicts EC numbers from removed ECs in training, possibly with more visual aids or detailed procedural descriptions.
Ensure clustering methods and test set generation are comprehensively described both in the text and supporting materials.